# Sodar Observation of the ABL Structure and Waves over the Black Sea Offshore Site

**Vasily Lyulyukin [1],\* , Margarita Kallistratova [1] , Daria Zaitseva [1], Dmitry Kuznetsov [1], Arseniy Artamonov [1] , Irina Repina [1,2] , Igor Petenko [1,3] and Rostislav Kouznetsov [1,4] and Artem Pashkin [1,2]**

1   A.M. Obukhov Institute of Atmospheric Physics, Russian Academy of Sciences, 119017 Moscow, Russia; margo@ifaran.ru (M.K.); zaycevadv@gmail.com (D.Z.); mikrer@yandex.ru (D.K.); sailer@ifaran.ru (A.A.); iar.ifaran@gmail.com (I.R.); i.petenko@isac.cnr.it (I.P.); rostislav.kouznetsov@fmi.fi (R.K.); artem.ifa64@gmail.com (A.P.)
2   Research Computer Center, M.V. Lomonosov Moscow State University, 119991 Moscow, Russia
3   Institute of Atmospheric Sciences and Climate, National Research Council, 00133 Rome, Italy
4   Finnish Meteorological Institute, FI-00101 Helsinki, Finland
\*   Correspondence: lyulyukin@ifaran.ru

**Abstract:** Sodar investigations of the breeze circulation and vertical structure of the atmospheric boundary layer (ABL) were carried out in the coastal zone of the Black Sea for ten days in June 2015. The measurements were preformed at a stationary oceanographic platform located 450 m from the southern coast of the Crimean Peninsula. Complex measurements of the ABL vertical structure were performed using the three-axis Doppler minisodar Latan-3m. Auxiliary measurements were provided by a temperature profiler and two automatic weather stations. During the campaign, the weather was mostly fair with a pronounced daily cycle. Characteristic features of breeze circulation in the studied area, primarily determined by the adjacent mountains, were revealed. Wave structures with amplitudes of up to 100 m were regularly observed by sodar over the sea surface. Various forms of Kelvin–Helmholtz billows, observed at the interface between the sea breeze and the return flow aloft, are described.

**Keywords:** breeze; sodar; atmospheric boundary layer; internal gravity waves; Kelvin–Helmholtz billows; Black Sea

## 1. Introduction

Over the past two decades, several hundred studies of sea breezes have been published. This has primarily been driven by an increased interest in wind energy, although regional weather events and the dispersion of air pollutants in coastal zones remain important issues as well. An overwhelming majority of the publications have been devoted to the numerical modeling of sea breezes (see, e.g., [1–7]). At the same time, the number of experimental studies of sea breezes in various countries and regions has also increased significantly. Thus, in the U.S.A., recommendations on the development of meteorology for coastal/offshore wind energy over the next 10 years were adopted in 2013 [8], which noted the need for "continuous, publicly available, multilevel measurements of winds and temperature over offshore waters".

A characteristic feature of experimental studies of sea breezes in recent years is the use of ground-based remote sensing as a supplement to the conventional measurements. Sounding using radar [9], sodar [10], and lidar [11] has made it possible to carry out studies of the vertical structure of sea breeze cells and fronts, encompassing return currents of the air. Ground-based remote sensing tools

have been used for more than half a century in atmospheric research, which have provided a wealth of knowledge about the structure and dynamics of the atmospheric boundary layer (ABL) in various locations and under various conditions [12]. Sodar is particularly suitable for studying the lower part of the ABL [13]. Sodar measurements, along with measurements of wind speed components, allow for the visualization of mesoscale turbulent structures, including internal gravity waves, and the determination of their parameters. Recently, with the help of sodar, wave motions in the stable boundary layer were studied in the mid-latitudes [14–17] and in Antarctica [18,19]; the main types of observed waves, as well as their periods and amplitudes, were determined. However, until now, in the research of sea breezes, sodar has been used mainly for wind-profiling, without registration of the inner mesoscale structures (see, e.g., [20–24]). Only in a few sodar breeze studies have examples of internal gravity waves in breeze density currents been shown [10,25].

At the same time, model simulations and laboratory experiments have indicated the complex mesoscale structure of breezes; in particular, the presence of Kelvin–Helmholtz billows (KHB) in the breeze front region and in the interfaces between the forward and reverse breeze currents [26–29]. Such wavy structures are of considerable interest. First, KHB cause a frictionlike force on the upper boundary of the air mass that slows the inland progression of sea breezes [26,30]. Thus, they can influence the exchange processes in the density currents, and therefore, should be taken into account in the numerical models. Secondly, wave movements and short-term bursts can directly affect the efficiency of wind turbines in offshore farms [31,32]. The above problems have stimulated the study of waves and mesoscale turbulent structures in sea breezes.

This paper presents the results of a study of sea breezes in the northern part of the Black Sea, held in June 2015 during a two-week expedition of the A.M. Obukhov Institute of Atmospheric Physics, Russian Academy of Sciences. The highly sensitive mini-sodar, Latan-3M, which was installed on an oceanological platform in the offshore zone of Crimea, served as the main measuring instrument. It is worth noting that, in recent years, great attention has been paid to the sea breezes in the coastal zones of the Black Sea [33–37] due to the promising development of wind power in this region which, so far, has only a small number of wind farms.

## 2. Measurement Site and Equipment

The studies were conducted at an oceanographic stationary platform in the coastal zone of the Black Sea in June 2015. The platform, managed by the Marine Hydrophysical Institute, is located on the shelf slope of the southern coast of the Crimea Peninsula (44.39° N, 33.99° E, Figure 1a). The location of the platform, at a distance of approximately 450 m from the coast and at a water depth about 30 m, makes it a unique observational point for collecting data in the coastal zone of an area that is usually lacking in remote sensing data sets. The coastline near the measurement site is significantly curved with a small bay to the north. Near the platform (in a 500 m vicinity), the coast extends from the southwest (SW) to the northeast (NE) direction; on a larger scale (up to 10 km), the coastline extends from west-southwest (WSW) to east-northeast (ENE). With this shape of the coast, the early sea breeze is expected to have a southeast (SE) direction, perpendicular to the nearest coast edge, then turning clockwise during the day due to enlargement of the breeze flow on the coast and the Coriolis force.

The described picture, however, does not take into account the topography of the coastal slope, which can have a critical effect on the local wind. The platform is located near a coast with a steep slope (with an average slope of about 200 m per 1 km distance) close to a plateau (Figure 1a). A topographic profile of the coast extending north from the platform is presented in Figure 1b. The presence of the steep coast could lead to the occurrence of katabatic and anabatic slope winds having the same direction as the breeze: during the day, from the coast up the slope (i.e., from sea to land), and at night, from the mountain down to the coast (i.e., from the coast to the sea, like the night breeze). This could, ideally, lead to a stronger breeze. A small glen runs north from the platform (from the coast to the edge of the plateau), which could help to establish a daily mountain–valley circulation, from the north at

night and from the south during the day. However, in general, the complex topography of the coast can also lead to the destruction of the ideal breeze circulation.

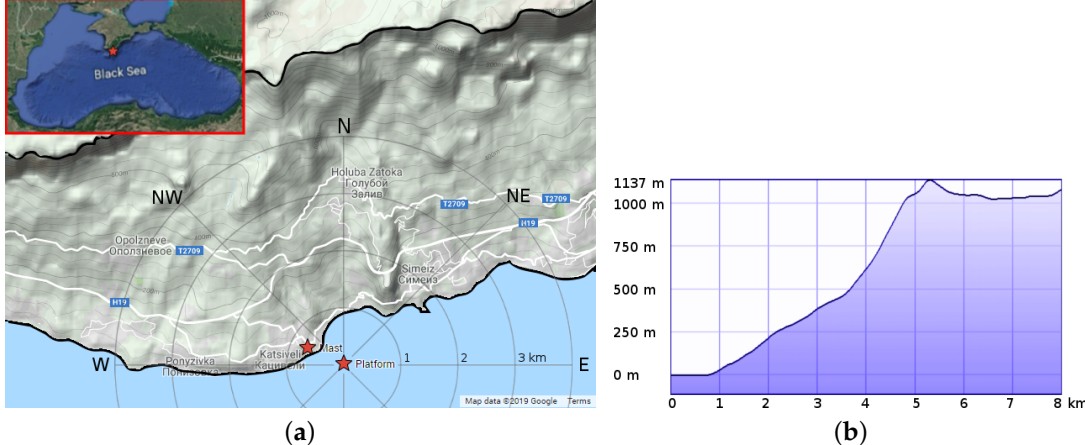

**Figure 1.** Experimental site location: (**a**) Location of the platform and the coastline topography at the area. The coastline and the edge of the plateau on the map are outlined with black lines. (**b**) Topographic profile of the coast extending north from the platform.

A general view of the platform is shown in Figure 2. In order to minimize the influence of the platform, the equipment was placed on the upper deck. We used the 3-beam Doppler minisodar Latan-3m, which was developed at the A.M. Obukhov Institute of Atmospheric Physics (Moscow, Russia) [38]. The sodar was operated with three 60 cm dish antennas at sounding frequencies of 3.3–4.4 kHz. The operating mode with frequency-coded sounding pulse [39] and parallel operation of the antennas was used to achieve a high temporal resolution. Different frequency coding was used to avoid cross-talk in the antennas: each antenna used an individual set of six frequencies emitted as a series of 50 ms pulses, which resulted in a vertical resolution of 10 m. The accuracy of the ABL parameter measurements with Latan-3 sodar was repeatedly verified by comparison with local measurements (see, e.g., [40]), and complied with generally accepted standards: $\pm 0.5$ m s$^{-1}$ for horizontal wind speed and $\pm 5$ degrees for wind direction. The antennas were mounted at 14 m above sea level (a.s.l.), one vertically pointed and two inclined and directed to the open sea. Accordingly, the lowest sounding level was 24 m above sea level. In Latan-3 sodars, the echo signal from each sounding is processed separately. Information of the instantaneous signal and noise intensities and the along-beam radial wind speed component are stored for each of the three antenna's range gates. During the campaign, the raw echo signals were stored to allow for further reprocessing, should it be necessary.

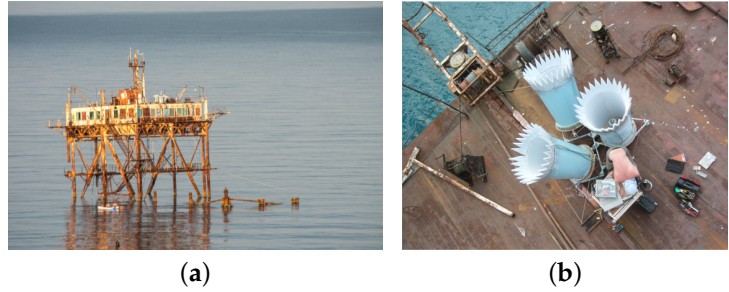

**Figure 2.** General view of the platform (**a**) and sodar during installation (**b**).

Standard meteorological measurements were provided by the WXT536 weather transmitter (Vaisala: Helsinki, Finland) placed on the upper deck of the platform on a small meteorological mast at level 15 m a.s.l. The vertical temperature profiles up to 600 m were measured by the

meteorological temperature profiler MTP-5 (Attex: Dolgoprudny, Russia) placed at 15 m a.s.l. and facing the open sea. The MTP-5 is an angular-scanning radiometer operating at 60 GHz, which provides data on the vertical temperature distribution with a height resolution of 50 m and a 5 min measurement cycle [41]. Two SMP21 pyranometers (280–3000 nm) and SGR3 pyrgeometers (4.4–50 μm) (Kipp&Zonen: Delft, The Netherlands) were operated on the platform to measure the downward and upward shortwave and longwave radiation.

Measurements of the air temperature, wind speed, and wind direction at the coast were carried out using the 150WX weather station (Airmar: Milford, USA), located on the meteorological mast on the roof of a building 100 m from the shore on the coastal slope. The height of the station above ground level (a.g.l.) was 10 m, and 45 m above sea level. The weather station was located 650 m to the west-northwest of the platform (Figure 1a).

## 3. Results

Sodar measurements on the platform were carried out in continuous mode from 12 to 22 June 2015, and related measurements were provided for most of this time period. The experiment was carried out mostly under fair weather conditions and a pronounced diurnal cycle of meteorological parameters was observed. Thermal stratification was determined by temperature profiles and dry adiabatic slope rate. Predominantly stable and neutral stratification was observed over the sea. Convective plumes were observed on sodar echograms for less than 20% of the time of measurement during offshore winds. The average relative humidity was about 70% and the maximum did not exceed 90%.

### 3.1. Diurnal Cycle of Meteorological Parameters

Figure 3 shows the time series of radiation budget, temperature, and wind speed and direction during the campaign. The periods of cloudy weather were determined from measurements of longwave and shortwave radiation (Figure 3a), according to Marty and Philipona [42]. Variations of the apparent emittance due to cloud cover significantly exceeded the variations of the clear sky emittance caused by variations of air humidity. We used the apparent emittance value as a criterion for determining cloudy periods. The threshold value of the emittance between clear sky and cloudy weather was chosen from the analysis of the time series of incoming shortwave radiation. In Figure 3, the identified time periods of cloudy weather are shaded (grey). The yellow bars indicate the local daytime periods. The cloudy days (12, 17, and 19 June) were excluded from the statistics. Figure 3b shows the time series of air temperature measured at the platform mast 15 m a.s.l., at the onshore mast 10 m a.g.l., and sea surface temperature (SST) obtained from radiometer measurements. Note that, during the entire experiment, the water temperature did not exceed the air temperature at 15 m a.s.l. Time series of the wind direction and speed from the data of sodar measurement and the onshore mast are given in Figure 3c,d, respectively. A steady west wind was observed daily during the daytime, with speed values of up to 12 m s$^{-1}$ at 50 m a.s.l. The night wind direction was less steady and generally ranged from the north-west to the east with typical speed values of about 2–3 m s$^{-1}$ at 50 m a.s.l. A rapid change in wind direction in the morning and evening hours was observed daily. The values of geostrophic wind speed and direction are also presented in Figure 3c,d, which were calculated from reanalysis data of sea level pressure by the United States National Center for Environmental Prediction (NCEP). During the experiment, the geostrophic wind direction was predominantly western and ranged from the northwest to the southeast. According to a quadrant classification (see, e.g., [43]), the geostrophic wind direction was generally from quadrants Q1 and Q3. The diurnal behavior of the probability density of the wind speed and direction, as well as the mean wind speed, is presented in Figure 4. The plots show a typical diurnal cycle of wind speed and direction, with dominant direction from the north for night hours (from 19:00 to 7:00 local time (+3 GMT)) and from the west (along the coast) for the daytime (from 7:00 to 19:00). The mean wind speed time course had two maxima: about 6 m s$^{-1}$ at around 03:00 and about 2.5 m s$^{-1}$ at around 15:00; and two minima: at 08:00 and at around midnight.

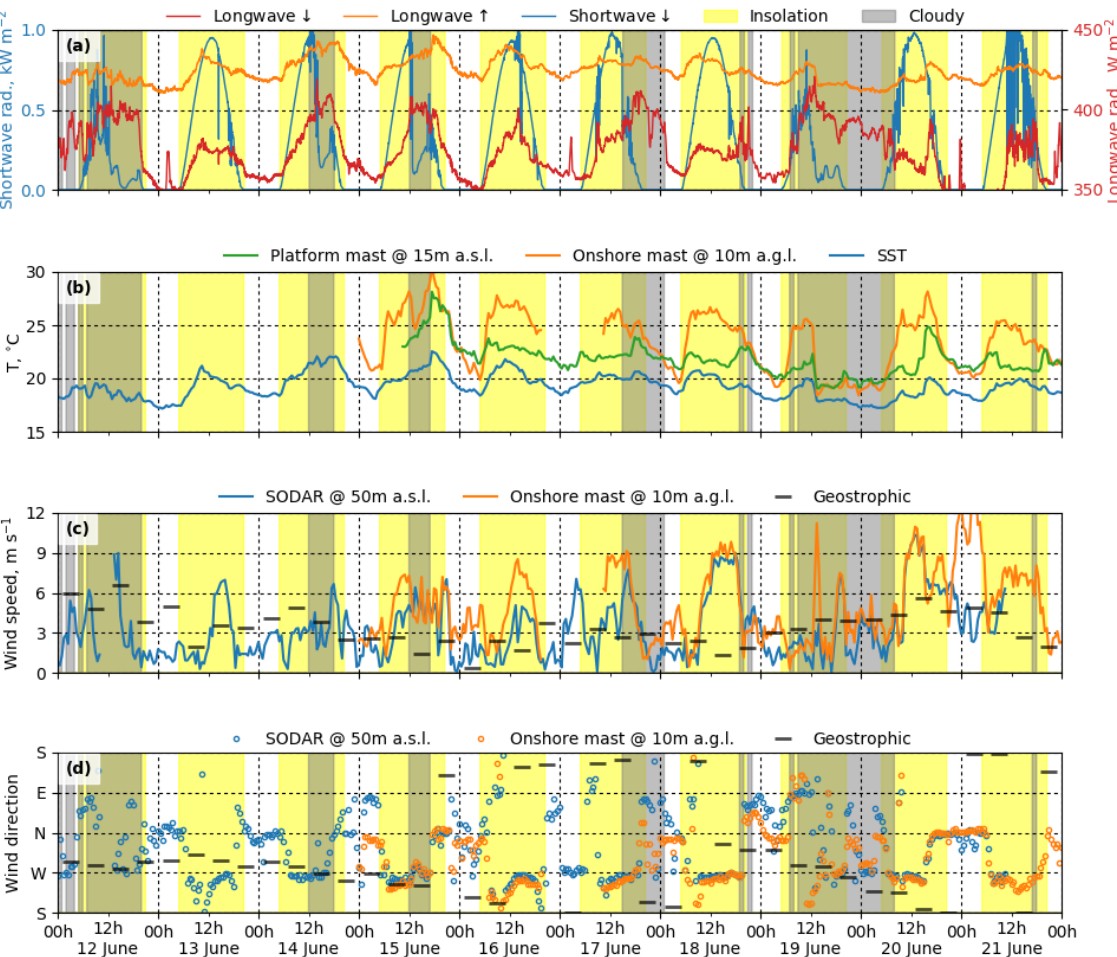

**Figure 3.** Time series of radiative fluxes (**a**), air and sea surface temperature (**b**), and wind speed and direction (**c**,**d**). The yellow bars indicate local daytime. Periods of cloudy weather are shaded.

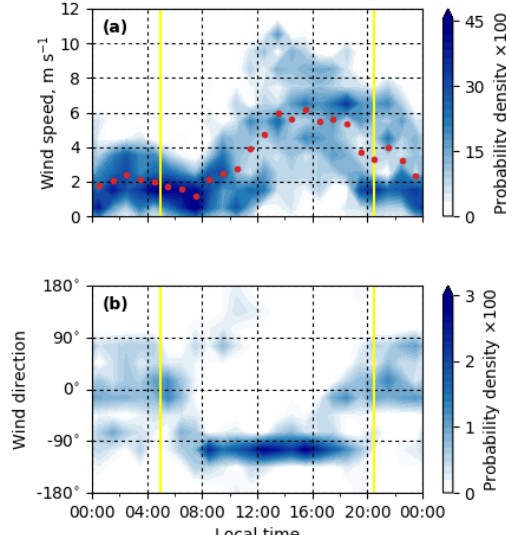

**Figure 4.** Diurnal behavior of the probability density of the wind speed (**a**) and wind direction (**b**) from the sodar at 50 m a.s.l. for days with fair weather. The red dots indicate the mean wind speed calculated for each time interval. Yellow lines indicate sunrise and sunset.

Wind roses from measurements at the platform and onshore masts for all days with fair weather are presented in Figure 5. The wind roses were built for both the entire time and separately for day and night hours, in accordance with the daily change of wind mode. The prevailing wind directions near sea level were from the west (along the coast) and from the north (in the direction of the coastal slope). A north wind was typical at night, whereas a west wind was typical during the day. Wind from the open sea (from the east) was observed sporadically in the morning hours and sometimes at night. Wind from the south was rare. The distribution of wind speed and direction, according to the measurements at the onshore mast, qualitatively repeated the distribution of winds at the platform; however, a slight shift in the wind direction is observable, which can be associated with the orography of the area.

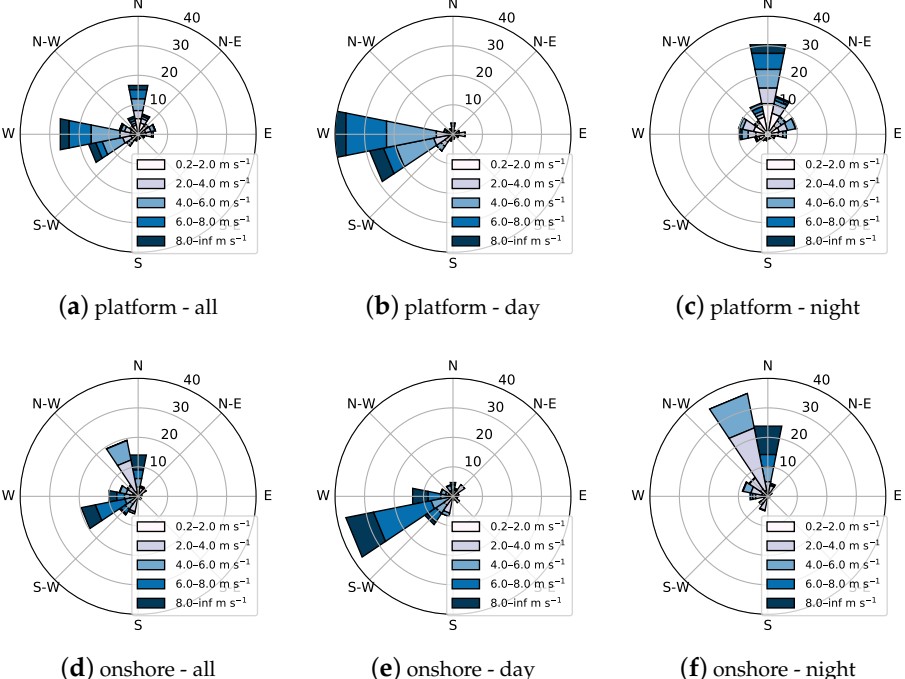

**Figure 5.** Wind roses near the surface for days with fair weather. (**a**–**c**) Wind from platform mast (15 m a.s.l.); (**d**–**f**) Wind from onshore mast (10 m a.g.l.). Left panels correspond to roses over the entire time; middle, for day hours (07:00–19:00 local time); and right, for night hours (19:00–07:00).

## 3.2. Vertical Structure of the Wind Field

Wind roses from sodar data for various heights are presented in Figure 6. Differences in the daytime and night-time distributions remained considerable with height, with western winds prevailing at all heights. The distribution of wind speed did not significantly vary with height; however, there were systematic changes in the distribution of wind directions. Western winds, which can be seen to be pronounced in the daytime wind roses, turned clockwise with increasing altitude (which corresponds to the effect of the Coriolis force). The night-time distributions corresponded to the situation with a return flow at heights above 100 m: the fraction of northern winds decreased with height and the fractions of winds with other directions increased. The daytime wind was more stable, and therefore, dominated in the whole-period distributions. The change with height was not pronounced, as one would expect, in the case of classical breeze circulation. Southern winds were rare at all heights. It is important to note that the sodar wind rose at 300 m is not fully representative, as the altitude range of sounding depends significantly on meteorological conditions, and wind speed data at 300 m were available for less than 30% of the time; mainly for winds from the shore.

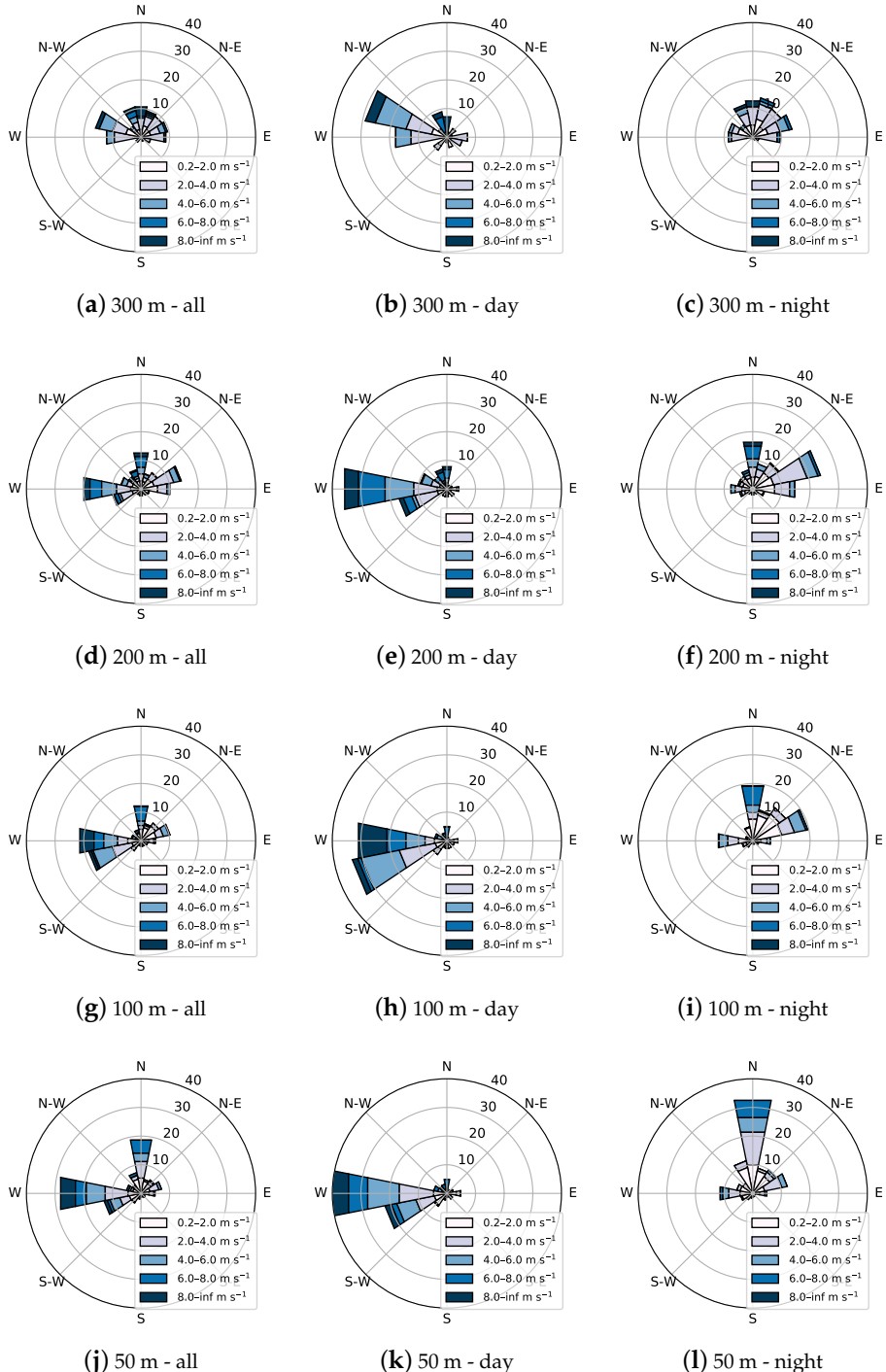

**Figure 6.** Windroses from sodar measurements for days with fair weather. (**a–c**) Wind at 300 m a.s.l. (**d–f**) Wind at 200 m a.s.l. (**g–i**) Wind at 100 m a.s.l. (**j–l**) Wind at 50 m a.s.l.

Figures 7 and 8 show examples of time series of vertical profiles of wind speed and direction, characterizing the typical vertical structure of the wind field. The profiles were calculated with half hour averaging periods. Examples of steady wind profiles are given in Figure 7: Figure 7a shows profiles for the situation of the northern wind near the sea surface at night. The wind direction changed from the north (offshore direction) to the east (onshore direction) with an increase in height. The minimum wind speed was located at an altitude of about 100 m. Both clockwise and counterclockwise (by 270 degrees) rotations are presented. The lower part of the profiles corresponds to the picture of a low-level jet

(LLJ) stream, with a maximum speed of about 3–4 m s$^{-1}$, located below the sounding range. Figure 7b shows an example of a steady west wind observed during the daytime. The wind direction was unchanged with height. The wind speed profile corresponds to a LLJ stream, with a maximum wind speed of up to 11 m s$^{-1}$, located below 50 m. In the absence of a steady western wind, frequent changes in the wind direction near the sea surface (by 90 degrees or more) were observed. In this case, as a rule, the change in wind direction did not occur simultaneously at different heights, forming transitional vertical profiles characterized by significant vertical wind shears. Figure 8 shows profile series for cases of change in the wind direction near the sea surface accompanied by a directional shear.

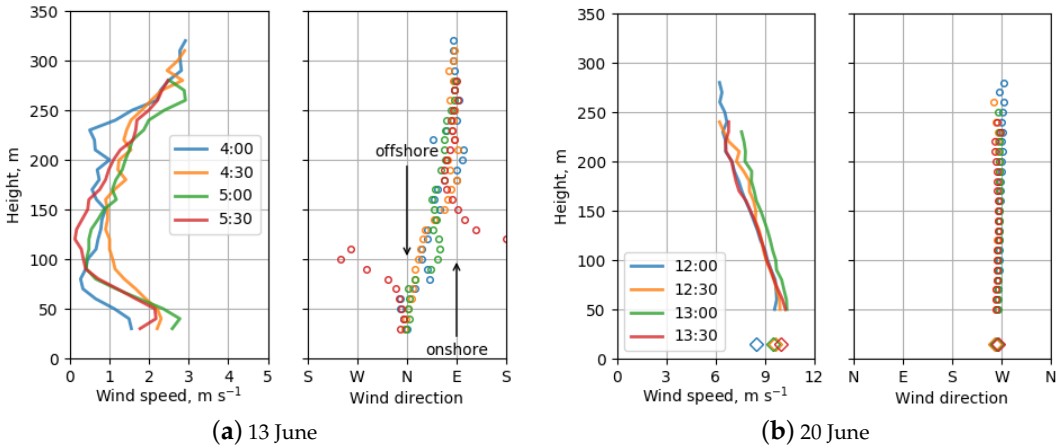

**Figure 7.** Half-hourly vertical profiles of wind speed and direction during steady winds from the north (**a**) and west (**b**). Diamonds indicate wind speed and direction measured at the platform mast at 15 m a.s.l.

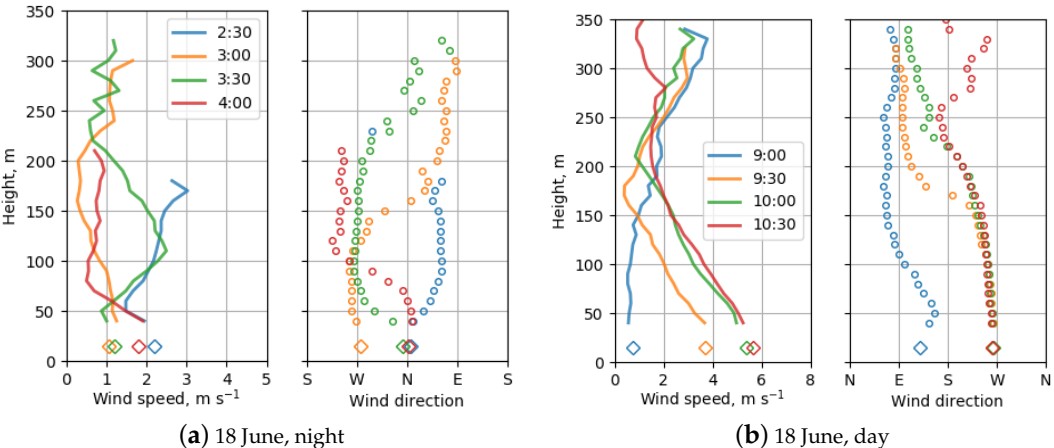

**Figure 8.** Vertical profiles of the wind speed and direction during transition periods.

Figure 9a shows the wind speed probability density distribution with height. The mean wind speed did not exceed 4 m s$^{-1}$ at all heights up to 300 m and had a local minimum at about 150 m a.s.l., which indicates the presence of breeze circulation. The histogram of the probability distribution of the wind speed at 50 m a.s.l. is given in Figure 9b.

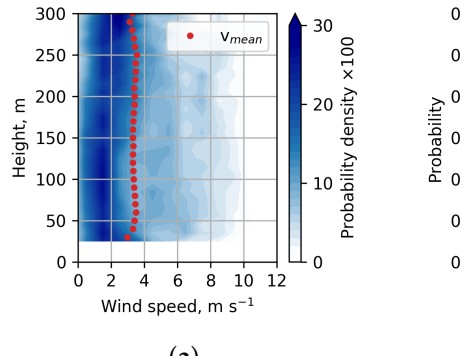
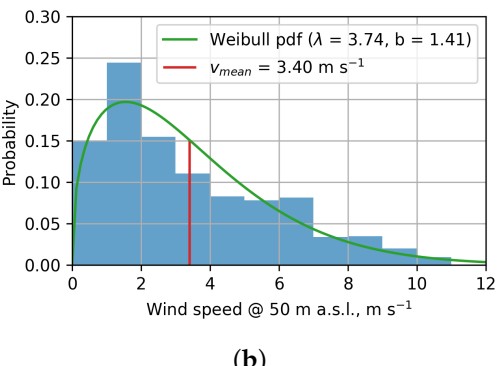

(**a**)    (**b**)

**Figure 9.** Wind speed probability density distribution with height (**a**) and a histogram of wind speed distribution at 50 m a.s.l. (**b**). The green line shows the Weibull distribution fitting.

### 3.3. Observation of Wave Structures in Shear Flows

Under stable ABL stratification, the presence of vertical wind shear in the layer can lead to Kelvin–Helmholtz instability and cause the formation of Kelvin–Helmholtz billows (KHB). During the campaign, such turbulent structures (in the form of braids or inclined stripes) were regularly observed in sodar echograms. The sodar return signal is proportional to the structural parameter $C_T^2$ [44], and therefore, can be considered as an indicator of turbulence. Figures 10 and 11 show examples of such observations: Figure 10 shows a series of wind speed and direction profiles, as well as sodar echograms for two episodes of observation of KHB in the nocturnal boundary layer under the north wind near the sea surface, changing to the east direction (onshore) with height. In the cases presented, in the layer of up to 100 m, the wind speed decreased with height (about 4 m s$^{-1}$ per 100 m) in the presence of LLJ with a maximum below the sounding range. In the field of the scattered signal in the lower turbulent layer, wave structures were observed in the form of billows tilted to the right. The temporal period of the observed structures was about 2 min, which is equal to (corresponding to the Taylor hypothesis of frozen turbulence) about 250 m in the spatial period. In the echogram in Figure 10a, KHB are observed in the layer of increasing wind speed above the level of the minimum wind speed, in the form of billows tilted to the left. The quasi period of the KHB was about 3–5 min (400–600 m) and the double amplitude of the wave (equal to thickness of the wavy layer) was about 150 m. Such shapes, with the opposite inclination of the billows, correspond to the typical KHB structures in shear flows with different signs of shear [45].

Kelvin–Helmholtz billows can lead to an increase in vertical heat and mass transfer due to the generation of turbulence [46]. Indirectly, the degree of vertical exchange can be judged by a vertical velocity field. The echograms with KHB structures and corresponding fields of vertical velocity obtained by the sodar are combined in Figure 11. A series of vertical profiles of wind speed and direction by sodar and temperature profile by the profiler are also presented in Figure 11. Dry adiabats are shown to estimate the temperature stratification. Figure 11a shows a matching for the case of wind direction changing from west to east with height. Two different wave layers can be observed. In the lower layer (up to 150 m), the wind speed decreased with height (about 2 m s$^{-1}$ per 100 m) and a wavy structure was observed as a series of quasi-periodic stripes, inclined to the right. In the upper layer (from 150–300 m) with increasing wind speed (about 2 m s$^{-1}$ per 100 m), the structure was observed as stripes tilted to the left. The temporal period of the observed structures was about 2 min, which is equal to a spatial period of about 250 m. In the vertical velocity fields presented, alternating areas of ascending and descending flows are visible; matching the shape of turbulent structures in the field of the sodar return signal. Figure 11b shows the case of a complex wave structure in several layers, with time periods of 1–3 min.

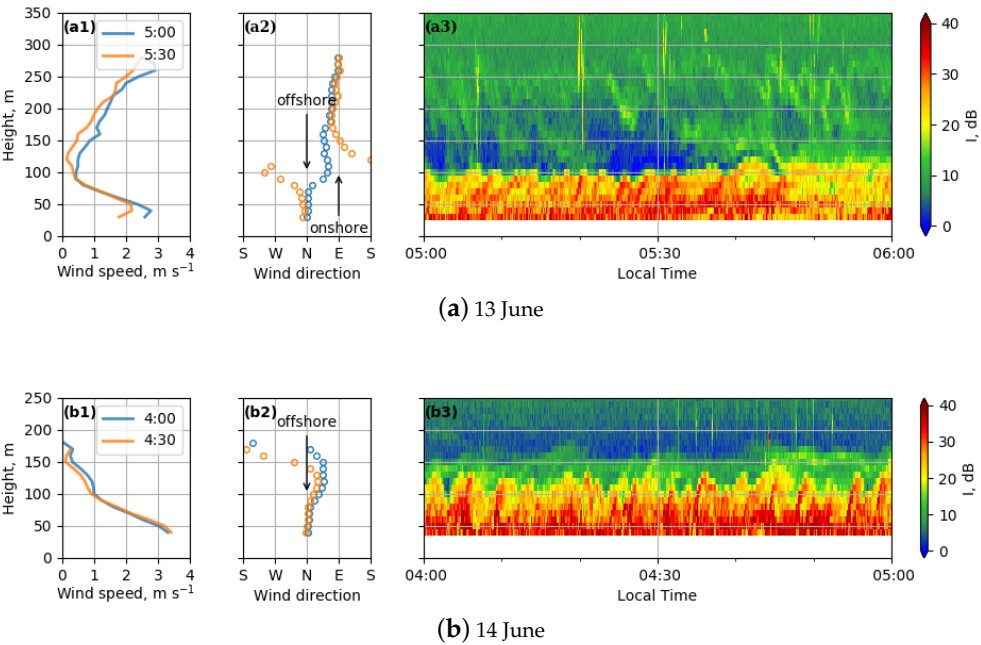

**Figure 10.** Two episodes of Kelvin–Helmholtz billows (KHB) observation in the nocturnal boundary layer under the north wind (offshore) near the sea surface and a return flow aloft (onshore) on the 13th (**a**) and 14th (**b**) of June. Panels (**a1,a2**) and (**b1,b2**) present half hourly vertical profiles of wind speed and direction, and panels (**a3,b3**) present the sodar return signal in height–time coordinates (echograms). The colors show the relative intensity of the return signal. Note the opposite orientation of the billows in the lower and upper parts of the echogram in episode (**a**).

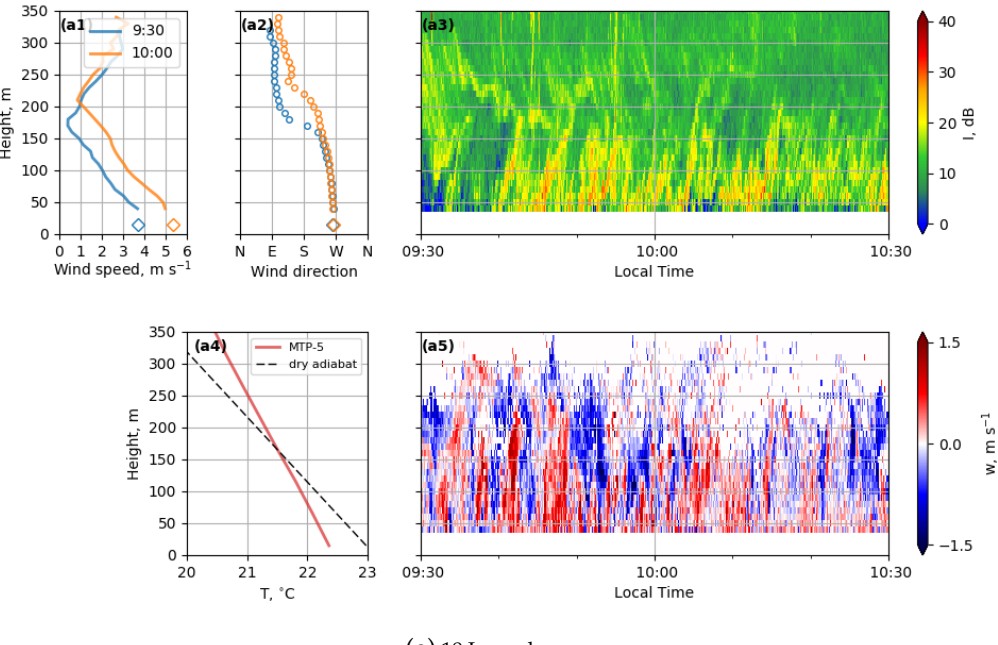

(**a**) 18 June, day

**Figure 11.** *Cont*.

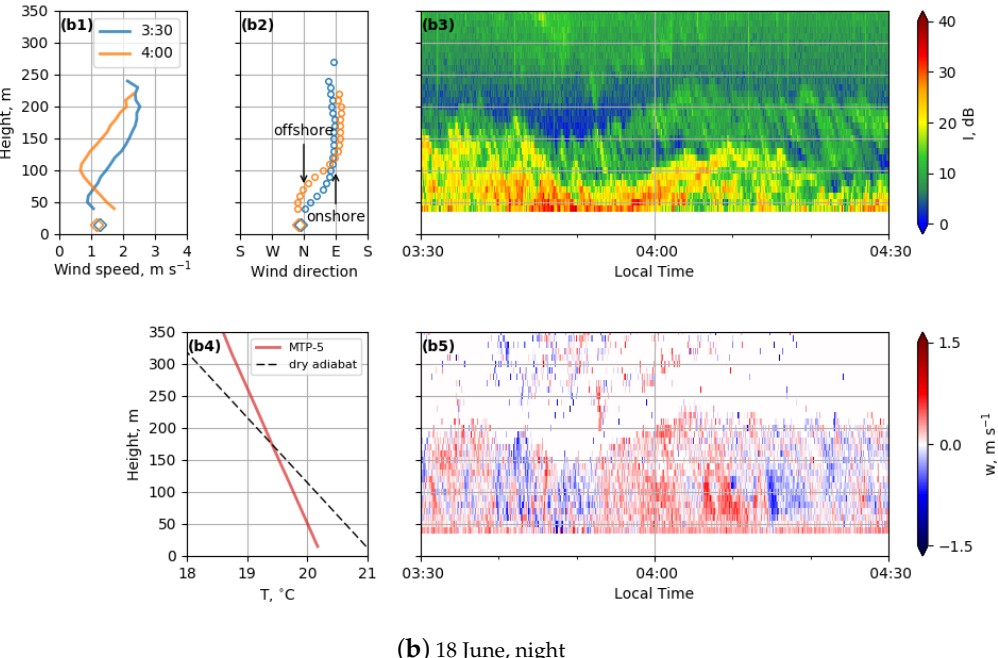

(**b**) 18 June, night

**Figure 11.** Examples of KHB episodes with vertical motion in the observed ABL. (**a**) Two different wave layers in the case of wind direction changing from west to east with height in the daytime. (**b**) A complex wave structure in several layers under the north wind (offshore) near the sea surface and a return flow aloft. Panels (**a4,b4**) show the temperature stratification by MTP-5 profiler with a one-hour averaging period. Panels (**a5,b5**) show the vertical velocity fields obtained by sodar.

Figure 12a shows echogram and vertical wind speed field for the case of strongly stable temperature stratification with the wind from the sea when the surface turbulent layer is below the sodar sounding range and vertical motions are not detected. Figure 12b represents the case of intense convection brought about by the north wind from the land under unstable temperature stratification.

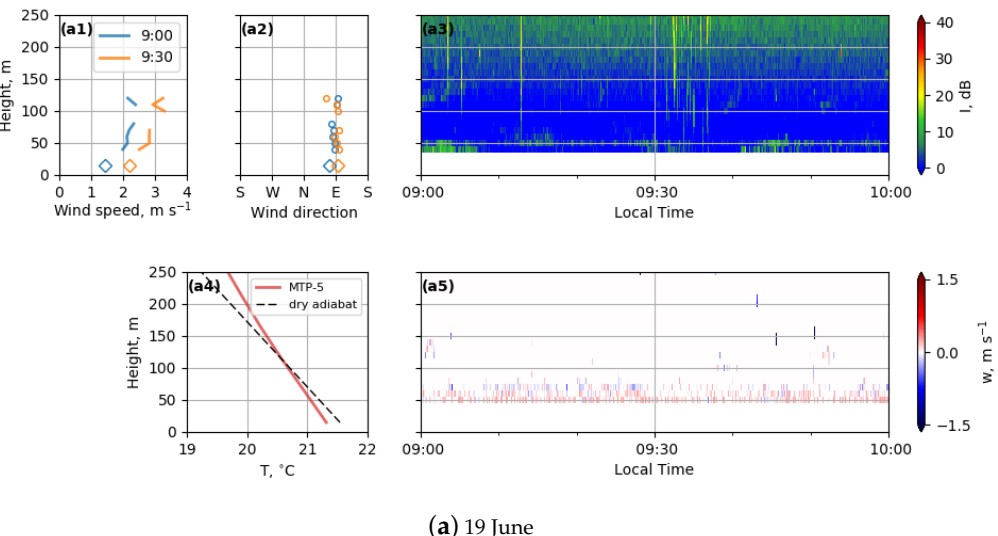

(**a**) 19 June

**Figure 12.** *Cont.*

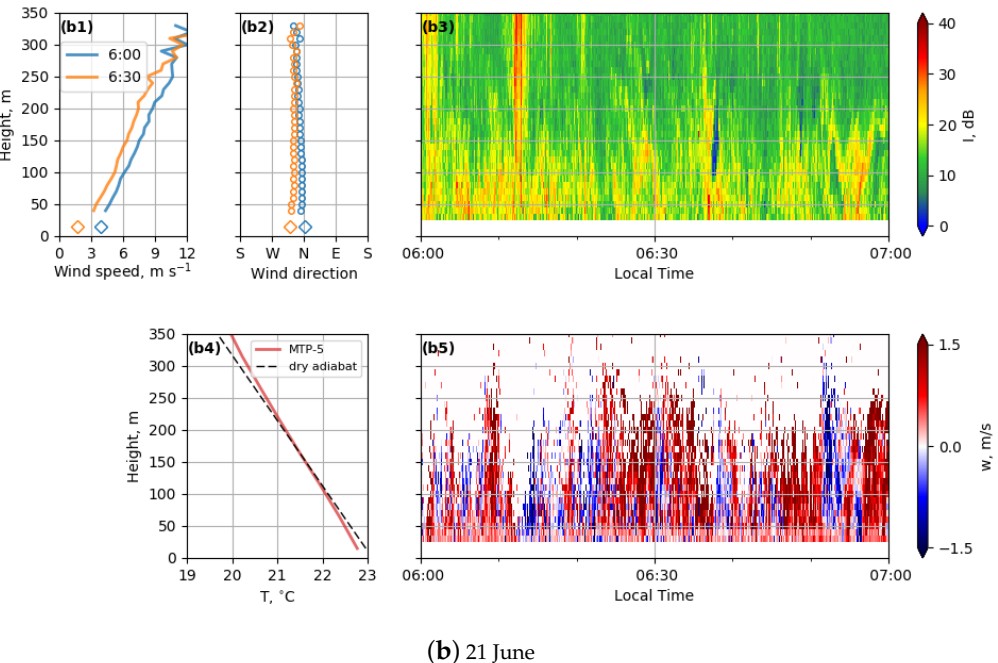

**(b)** 21 June

**Figure 12.** Episodes representing strongly stable temperature stratification with the wind from the sea (**a**) and convection under unstable temperature stratification with the wind from the shore (**b**).

## 4. Discussion

During the experiment, a pronounced diurnal cycle in the wind field was observed, which was accompanied by various sporadic changes in wind direction near the sea surface; however, breeze circulation in the "pure" form was not observed. Cases of rapid change in wind direction with a height similar to the breeze return flow pattern were observed rarely. Usually, such episodes lasted no more than 3 h and the wind direction in the return flow was unstable. A steady west wind was observed during the daytime (and sometimes at night; for example, 17 June from 00:00–07:00 local time), which changed direction slightly and had a maximum speed of 12 m s$^{-1}$ (at 50 m a.s.l.), which is difficult to explain with only pure breeze circulation. Most likely, multiple factors acted simultaneously; such as thermal pressure gradient, synoptic wind, and the orographic features of the coast. When the geostrophic wind direction was from quadrants Q1 or Q3, the daytime thermal pressure gradient should lead to an increase in the along-shore component of the near-surface wind (from west (W) to west-southwest (SWW) in our case). Geostrophic wind from Q1 was also favorable for the development of a "corkscrew" sea breeze [26,47]. Nocturnal offshore wind flow accompanied by a change in direction to the east (from the open sea sector) at an altitude of about 200 m could be conditionally attributed to enhancement of the night breeze by the katabatic flow. The nocturnal wind flow was less steady and noticeably weaker than the daytime wind, and had a typical speed of about 2–3 m s$^{-1}$ at 50 m a.s.l. Sometimes, wind from the east was observed at the platform, while a north wind was observed at the onshore mast.

A comparison of the wind direction and speed in the lower part of the ABL with wind data for this site from the literature [33,48] verifies that the observed pattern of wind distribution was quite representative for this area in the summer season. Numerical modeling of the breeze for this region [35] gives average wind speed values of about 1–2 m s$^{-1}$ with SW and NE directions for the day and night breezes, respectively, at time of maximum development in the summer season. The simulated wind velocity fields demonstrate that, in the region of the Southern coast of the Crimean Peninsula, the day and night breezes over the sea are primarily determined by small-scale inhomogeneities of the coast and by the adjacent mountains. The day breeze is blocked by the (fairly high) Crimean mountains, and

the night breeze has the character of katabatic flow from the mountain range. It has also been shown that the night breeze is weaker than the day breeze due to lower temperature gradients. However, the spatial resolution of the modeling results presented in [35] was too low for a detailed comparison with the observational data.

The shape and structure of the KHB observed in the shear flows during the experiment coincided with the previous studies of KHB described by the authors [14,49]. The periods of the waves were consistent with the results of sodar observations of breezes in [10]. The cases of simultaneous observation of two wave layers with different inclinations of KHB was similar to those described in [50], but with the opposite orientation of the tilts. Note that the KHB were observed in the form of short (merely several periods of wave) trains, in contrast to KHB in low-level jet streams over a uniform land, where KHB trains of several hours have been observed [15]. This corresponds to the overall strong and rapid variability of the mesoscale sea breeze pattern recorded in our experiment.

Strong directional shear has a significant effect on the stability of stratification in the layer due to shear instability. The wide variety of observed wind speeds and direction profiles, along with their frequent and unpredictable changes, made it difficult to predict dynamical stability conditions. Statistical analysis of long time series of remote sensing observations, organized with regard to local features of the coastline, is necessary for their description. To assess the effect of synoptic conditions, analysis of long time series of continuous observations is also necessary.

## 5. Conclusions

For the first time, sodar measurements of ABL parameters were carried out in the coastal zone of the northern part of the Black Sea. During 10 days in June 2015, a three-axis Doppler sodar was operated on a stationary oceanographic platform located at a distance of 450 m from the coastline. This made it possible to investigate vertical profiles of wind speed and direction, as well as mesoscale wave structures in the sea breeze flows.

The observed diurnal cycle of the mean wind speed had two maxima: 6 m s$^{-1}$ at around 03:00 and 2.5 m s$^{-1}$ at around 15:00. A steady west wind was observed in the daytime, with speed values of up to 12 m s$^{-1}$ at 50 m a.s.l. The night wind was less steady and noticeably weaker, with typical speed values of about 2–3 m s$^{-1}$ at 50 m a.s.l. A rapid change of wind direction in the morning and evening hours was observed daily. Return air flow aloft was rarely observed at altitudes of 200–300 m. A characteristic feature of the winds in the studied area was a typical difference of about 90° between night and day near-surface flows. The main direction of the daytime wind at all heights was from the SW–W sector. The night wind did not blow from the opposite direction, but from the N–NE sector; and rarely from the NW–N sector. The night wind was mainly determined by the katabatic air flow from the slopes of the coastal mountain range.

Sodar echograms revealed many episodes of wave activity in the ABL over the coastal zone. Short trains of Kelvin–Helmholtz billows, in the form of braids, were observed, usually at dawn in the upper part of the onshore sea breeze flow (i.e., at altitudes where the wind speed fell with altitude). The KHB periods were 2–4 min. In some cases, KHB with periods of 7–8 min were also observed in the lower part of the return flow, in a layer of decreasing wind speed. The inclinations of the billows in the lower and upper flows were opposite.

On the basis of our short-term measurements, the studied area seems to be unsuitable for wind energy use due to the low mean wind speed, its strong variability, and the strong intermittency of mesoscale and wave structures in the turbulence and wind speed. However, for comprehensive inference, long-term experimental studies in different seasons are needed.

**Author Contributions:** Conceptualization, V.L., M.K., I.R., and I.P.; Data curation, V.L., D.Z., D.K., and I.R.; Formal analysis, V.L., D.Z., A.A., and A.P.; Funding acquisition, M.K. and I.R.; Investigation, V.L., D.K., and A.A.; Methodology, V.L., M.K., and R.K.; Project administration, I.R.; Resources, V.L., D.K., A.A., I.R., and R.K.; Software, V.L. and R.K.; Supervision, M.K.; Validation, V.L., D.Z., and R.K.; Visualization, V.L. and D.Z.; Writing—original draft preparation, V.L.; Writing—review and editing, M.K. and I.P.

**Funding:** This research was funded by the Russian Foundation for Basic Research through grants 19-05-01008 and 19-05-00547; organization of the expedition, and primary processing and analysis of measurement data were supported by the Russian Science Foundation through grants 14-27-00134 and 17-17-01210.

**Acknowledgments:** The authors are grateful to Otto Chkhetiani for help in organizing the expedition, Valerii Kramar for data of measurements by the weather station at the onshore mast, and Mikhail Varentsov for the photo of the platform. NCEP Reanalysis data was provided by the NOAA/OAR/ESRL PSD, Boulder, Colorado, USA, from their Web site at https://www.esrl.noaa.gov/psd/. The primary measurement data from the current study can be obtained from the first co-author of the article upon request.

**Conflicts of Interest:** The authors declare no competing interests. The funders had no role in the design of the study; in the collection, analyses, or interpretation of data; in the writing of the manuscript, or in the decision to publish the results.

## Abbreviations

The following abbreviations are used in this manuscript:

ABL  atmospheric boundary layer
a.s.l.  above sea level
a.g.l.  above ground level
KHB  Kelvin-Helmholtz billows
LLJ  low-level jet
SST  sea surface temperature

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
