# Peer review of "Sodar Observation of the ABL Structure and Waves over the Black Sea Offshore Site"

_atmosphere, doi:10.3390/atmos10120811_

Round 1

Reviewer 2 Report

This manuscript is interesting and suitable for publication in this journal.

Among the advantages of the paper is its experimental character and abundance of experimental data.

However, the work is illustrative in character, it comprises many experimental results as plots, but the accuracy of the data and their statistical characteristics such as variances and confidence intervals are not discussed at all in the manuscript. 

In my opinion, Fig. 1 a should be eliminated, because Fig. 1 c provides the same information in more detail.

Two plots on the upper left of Fig. 11 a have many empty space and should be re-drawn. The echogram on the right at the bottom of Fig. 11 a is simply a chaotic set of points.

Many figures illustrate the wind rose and are not discussed and somehow analyzed and generalized in the text. The goal of their presentation is not clear.

References 29-31 are absent in the text of the manuscript.

The language should also be corrected (fluxes density, minutes averaging, braids (?) or inclined strips (?) in echograms, abbreviation JUN, daily in the daytime, coastal strip, etc.). 

The conclusions are poorly justified due to the absence of statistical analysis of the numerical data obtained.

The manuscript neads major revision.

In its present form, I cannot recommend this manuscript for publication. 

Round 2

Reviewer 2 Report

The manuscript in its present form can be published.